# The Loading of Epigallocatechin Gallate on Bovine Serum Albumin and Pullulan-Based Nanoparticles as Effective Antioxidant

**DOI:** 10.3390/foods11244074

**Published:** 2022-12-16

**Authors:** Zikun Li, Xiaohan Wang, Man Zhang, Hongjun He, Bin Liang, Chanchan Sun, Xiulian Li, Changjian Ji

**Affiliations:** 1College of Life Sciences, Yantai University, Yantai 264005, China; 2College of Food Engineering, Ludong University, Yantai 264025, China; 3School of Pharmacy, Binzhou Medical University, Yantai 264003, China; 4Department of Physics and Electronic Engineering, Qilu Normal University, Jinan 250200, China

**Keywords:** bovine serum albumin, pullulan, epigallocatechin gallate, nanoparticles, antioxidant activity

## Abstract

Due to its poor stability and rapid metabolism, the biological activity and absorption of epigallocatechin gallate (EGCG) is limited. In this work, EGCG-loaded bovine serum albumin (BSA)/pullulan (PUL) nanoparticles (BPENs) were successfully fabricated via self-assembly. This assembly was driven by hydrogen bonding, which provided the desired EGCG loading efficiency, high stability, and a strong antioxidant capacity. The encapsulation efficiency of the BPENs was above 99.0%. BPENs have high antioxidant activity in vitro, and, in this study, their antioxidant capacity increased with an increase in the EGCG concentration. The in vitro release assays showed that the BPENs were released continuously over 6 h. The Fourier transform infrared spectra (FTIR) analysis indicated the presence of hydrogen bonding, hydrophobic interactions, and electrostatic interactions, which were the driving forces for the formation of the EGCG carrier nanoparticles. Furthermore, the transmission electron microscope (TEM) images demonstrated that the BSA/PUL-based nanoparticles (BPNs) and BPENs both exhibited regular spherical particles. In conclusion, BPENs are good delivery carriers for enhancing the stability and antioxidant activity of EGCG.

## 1. Introduction

Previous studies have shown that catechins have antioxidant effects [1] and that they are highly effective natural antioxidants with low toxicity [2]. Furthermore, they also have anticancer [3], anti-obesity [4], and anti-inflammatory properties [5]. Catechins contain three basic ring nuclei—A, B and C—that form a D ring after esterification. According to the different groups connected to the B and C rings, common catechins can be divided into four forms, which can be divided as follows: epicatechin (EC), epigallocatechin (EGC), epicatechin gallate (ECG), and epigallocatechin gallate (EGCG) [6]. Among them, EGCG has the highest content, accounting for approximately 70% of the total catechins. The molecular structure of EGCG contains eight monophenolic hydroxyl groups and it is the most active catechin.

Although EGCG has various beneficial health effects, its application in pharmaceuticals and nutritional supplements is very limited, mainly due to its low bioavailability [7]. In addition, reactions such as oxidation and polymerization can occur easily under conditions such as high temperatures and high humidity, and it can be easily decomposed in neutral and alkaline physiological environments [8]. The hydrophobic benzene ring and more than five hydrophilic hydroxyl groups in the molecule make EGCG both hydrophilic and lipophilic. However, the solubility of EGCG is poor. Its low solubility and poor stability lead to its low bioavailability [9]; however, these problems can be effectively solved by nano-delivery systems.

Nanoparticles are fine microparticles with more than one dimension, and they are between 1–100 nm in size. Furthermore, they are generally composed of natural macromolecular substances or synthetic macromolecular substances. They can be used as carriers for conducting or delivering active substances and drugs. Embedding small-molecule active substances or drugs in nanoparticles can alter their release rate, increase the permeability of their biofilms, and improve their bioavailability [10].

The use of bovine serum albumin (BSA) to prepare nanoparticles as carriers has been gradually gaining attention. Due to the special heart-shaped structure of BSA, a large number of hydrophobic amino acids are concentrated inside it, which form a hydrophobic inner core and a hydrophilic outer shell. This structure allows BSA to interact with both hydrophilic and hydrophobic small molecules [11]. BSA is widely used as a drug carrier due to its non-antigenicity, its degradability, and its nontoxicity [12], in spite of its allergenic disadvantage [13].

Pullulan (PUL) is a natural, degradable macromolecular polymer that is nonhazardous and has been widely used in many fields [14]. PUL contains three hydroxyl groups in each glucose unit and has good solubility in aqueous solutions. Its aqueous solution is stable and less dense than other water-soluble polysaccharides and is not affected by temperature, pH, or most metal ions. Thus, PUL is often used as a food additive [15]. It is widely used in food processing because of its high non-immunogenicity [16], non-carcinogenicity [17], and biocompatibility [18].

In this context, an attempt was made to encapsulate EGCG into a BSA-PUL nanoparticulate (BPNs) system to retain or enhance its antioxidant potential. The objectives of this study were as follows: (1) to investigate the particle size, the polydispersity index (PDI), zeta potential, and thermal stability of BSA or BSA-PUL complexes prepared at different ratios of BSA and PUL or with different pH solutions; (2) to study the encapsulation efficiency and loading rate of EGCG-loaded BPNs (BPENs); (3) to determine the effect of EGCG concentrations on the antioxidant activity and to release the capacity of BPENs in vitro; and (4) to characterize the structures of BPNs and BPENs using Fourier transform infrared spectra (FTIR) and a transmission electron microscope (TEM).

## 2. Materials and Methods

### 2.1. Materials

Bovine serum albumin (BSA), epigallocatechin gallate (EGCG), anhydrous ferric chloride, potassium ferricyanide, ferrous ammonium sulfate hexahydrate, salicylic acid, trichloroacetic acid, 1,1-diphenyl-2-Picrylhydrazine (DPPH), and 2,2′-Diaza-bis-3-ethylbenzothiazoline-6-sulfonic acid (ABTS) were obtained from Shanghai Macklin Biochemical Co., Ltd. (Shanghai, China). Pullulan (PUL) was purchased from Shanghai Yien Chemical Technology Co., Ltd. (Shanghai, China). Hydrogen peroxide (purity ≥ 30%), potassium dihydrogen phosphate, dipotassium hydrogen phosphate, absolute ethanol, hydrochloric acid, and sodium hydroxide were purchased from China National Medicines Corporation Ltd. (Shanghai, China).

### 2.2. Preparation and Characterization of BPNs

#### 2.2.1. Preparation of BPNs

First, 0.25 g of BSA and 0.25 g of PUL were dissolved in 50 mL of deionized water. After complete dissolution, the solution was placed in a refrigerator at 4 °C overnight to fully hydrate.

The 5 mg/mL PUL stock solution was diluted to the following three concentrations: 2.5 mg/mL, 1 mg/mL, and 0.5 mg/mL. Equal volumes of 5 mg/mL BSA solution and four concentrations of PUL (5 mg/mL, 2.5 mg/mL, 1 mg/mL, and 0.5 mg/mL) were mixed. Finally, the ratios of BSA/PUL were 1:1, 2:1, 5:1, and 10:1 (all of the BSA concentrations were 2.5 mg/mL). The mixtures of each ratio were adjusted to pH 4.5, 5.0, 5.5, 6.0, 6.5, and 7.0, with 0.01 M hydrochloric acid and 0.01 M sodium hydroxide solution. Next, the solution was incubated overnight to enable it to become completely mixed and homogeneous. The preparation of BPNs is shown in Figure 1(1).

Samples with different BSA-PUL ratios were called BPNs (1:1), BPNs (2:1), BPNs (5:1), and BPNs (10:1).

#### 2.2.2. Determination of Particle Size, PDI, and Zeta-Potential of BPNs

The particle size and distribution of the samples were determined with a particle size analyzer (90Plus Zeta, BIC, NYC, USA). All determinations were performed using three batches of the samples, and the results are expressed as mean ± standard error.

#### 2.2.3. Thermal Stability

The BSA and BPNs solutions with different ratios and pH levels were heated at 80 °C for 30 min and then cooled to room temperature in ice water [19]. The turbidity (OD_600_) of the BSA and BPNs solutions with and without a heating treatment was measured using a UV spectrophotometer (UV-2500, Shimadzu, Shanghai, China). The turbidity curves were plotted with the change in turbidity (OD_600_) as the dependent variable and the change in pH as the independent variable to characterize the thermal stability of the BSA and BPNs.

### 2.3. Preparation and Characterization of EGCG-Loaded BPNs (BPENs)

#### 2.3.1. Preparation of BPENs

Different concentrations (0.5 mg/mL, 1.0 mg/mL, 1.5 mg/mL, 2.0 mg/mL, and 2.5 mg/mL) of EGCG solutions were added to the BPNs solution, which was prepared under optimum conditions. The EGCG solution was mixed thoroughly with the protein solution, and then it was mixed with the polysaccharide solution. After adjusting the dispersion to the desired pH, using a 0.1 M hydrochloric acid solution or sodium hydroxide solution, the samples were left overnight and lyophilized. The preparation of BPENs is shown in Figure 1(2).

In this paper, the samples loaded with different concentrations of EGCG are called BPENs (0.5), BPENs (1.0), BPENs (1.5), BPENs (2.0), and BPENs (2.5).

#### 2.3.2. Encapsulation Efficiency and Loading Rate

The BPEN solutions were centrifuged at 3500× *g* for 30 min. The free EGCG in the clear supernatant was measured according to the vanillin–hydrochloric acid colorimetric method. The measurement of EGCG was determined by the absorbance method at 285 nm, and the standard curve is presented in Appendix A. All analyses were performed using three batches of the samples. The results were calculated according to Equations (1) and (2) [20]:(1)$$\mathrm{Encapsulation}\;\mathrm{efficiency}\;\left(\%\right)=\left(1-\frac{\mathrm{The}\;\mathrm{amount}\;\mathrm{of}\;\mathrm{free}\;\mathrm{EGCG}}{\;\mathrm{The}\;\mathrm{amount}\;\mathrm{of}\;\mathrm{total}\;\mathrm{EGCG}}\right)\times 100\%$$
(2)$$\mathrm{Loading}\;\mathrm{rate}\;\left(\mathrm{mg}\;\mathrm{EGCG}/\mathrm{mg}\;\mathrm{BSA}\right)=\frac{\mathrm{Encapsulated}\;\mathrm{EGCG}\;\mathrm{amount}}{\mathrm{Total}\;\mathrm{BSA}\;\mathrm{amount}}$$

#### 2.3.3. DPPH Scavenging Assay

We modified this assay slightly according to the method used by Tang et al. [21]. We added 2 mL of the sample solutions to the 2 mL of 0.2 mM DPPH% ethanol solution. The reaction mixture was incubated for 30 min on a shaker at room temperature in a dark place. The absorbance of the reaction solution was recorded at 517 nm, using a UV spectrophotometer (UV-2500, Shimadzu, Shanghai, China). The DPPH scavenging activity (%) was calculated using Equation (3), as follows:(3)$$\mathrm{DPPH}\;\mathrm{scavenging}\;\mathrm{activity}\;\left(\%\right)=\frac{\left(A_{0}-A_{i}\right)}{A_{0}}\times 100\%$$
where A_0_ is the absorbance of the DPPH solution after reacting with deionized water, and A_i_ is the absorbance of the DPPH solution after reacting with the sample.

#### 2.3.4. ABTS^+^ Scavenging Assay

The method was slightly changed with reference to Wang et al. [22]. ABTS powder was dissolved in deionized water to prepare a 7 mM ABTS solution. Potassium persulfate was dissolved in a 75 mM phosphate buffer solution (PBS) at pH 7.4 to prepare a 4.9 mM potassium persulfate solution. Next, ABTS^+^ was generated by the equal volume oxidation of the ABTS solution and the aforementioned potassium persulfate solution. Next, the ABTS^+^ solution was diluted with PBS to the absorbance value of 0.70 ± 0.02 (λ = 734 nm). Finally, 0.2 mL of the sample solutions were added to 4 mL of the ABTS^+^ solution and mixed thoroughly. Immediately after incubation in the dark for 15 min, the absorbance of the reaction mixture was recorded at 734 nm. The ABTS^+^ scavenging activity was calculated using Equation (4), as follows:(4)$${\mathrm{ABTS}}^{+}\;\mathrm{scavenging}\;\mathrm{assay}\;\left(\%\right)=\frac{A_{0}-{\;A}_{i}}{A_{0}}\times 100\%$$
where A_0_ represents the absorbance value of the ABTS^+^ solution after reacting with deionized water, and A_i_ represents the absorbance of the ABTS^+^ solution after reacting with the samples.

#### 2.3.5. Ferric-Reducing Antioxidant Power (FRAP) Assay

This experiment was carried out with reference to the method of Liu et al. [23]. We added 30 μL of the BPENs solution and 2.5 mL of 10 g/L potassium ferricyanide to 2.5 mL of PBS (0.2 mol/L, pH 6.6). The mixed samples were put into thermostatic water for 15 min at 37 °C. After adding 2.5 mL of 100 g/L trichloroacetic acid, 2.5 mL of the supernatant was taken after standing. Next, 2.5 mL of deionized water and 0.5 mL of 1 g/L FeCl_3_ were added to the supernatant, and, subsequently, the absorbance was measured at 700 nm.

#### 2.3.6. Hydroxyl Radicals (OH) Scavenging Assay

The hydroxyl radical (OH) was generated by a Fenton reaction model system, and the scavenging activity of the nanoparticles was determined following the procedures of Yang et al. [24]. We mixed 1.0 mL of 9 mmol/L ammonium ferrous sulfate and 1.0 mL of a 9 mmol/L salicylic acid–ethanol solution and left the solution for 10 min. We added 1.0 mL of an 8.8 mmol/L H_2_O_2_ solution and 1.0 mL of the BPENs solutions to the above solution. After mixing evenly, the mixture was placed at 37 °C to react for 1 h. Subsequently, the absorbance was measured at 510 nm. The ·OH scavenging activity was calculated using Equation (5), as follows:(5)$$\mathrm{Hydroxyl}\;\mathrm{radical}\;\mathrm{scavenging}\;\mathrm{rate}\;\left(\%\right)=A_{0}-\frac{A_{i}-{\;A}_{i0}}{A_{0}}\times 100\%$$
where A_0_ is the absorbance of the ·OH solution after reacting with deionized water, A_i_ is the absorbance of the ·OH solution after reacting with the BPENs solutions, and A_i0_ represents the absorbance of the ·OH solution after reacting with the BPENs solutions, without H_2_O_2_.

#### 2.3.7. In Vitro Release Behavior

The in vitro release behaviors of BPENs with different EGCG contents were studied using the dynamic dialysis method in deionized water [25]. Briefly, the dispersion of BPENs in distilled water was placed into dialysis tubing (molecular weight cut-off 8–14 kDa, Solarbio Science and Technology Co., Ltd., Beijing, China) and dialyzed against deionized water at 37 ± 0.2 °C in an air-bath shaker at 120 rpm. At predefined time intervals, 4 mL of the release media were collected, and the fresh release media were added. The cumulative percentage of EGCG released was determined at 285 nm and calculated using Equation (6), as follows:(6)$$\mathrm{The}\;\mathrm{cumulative}\;\mathrm{percentage}\;\mathrm{of}\;\mathrm{EGCG}\;\mathrm{released}=\frac{\sum_{i=1}^{13}C_{i}\left[V_{2}-\left(i\;-1\right)V_{1}\right]}{W}$$
where C_i_ is the concentration of EGCG in the released medium obtained in time i, mg/mL; V_1_ is the volume of the release media, 4 mL; and V_2_ is the volume of dialysis deionized water, 200 mL.

#### 2.3.8. Fourier Transform Infrared Spectra (FTIR) Analysis

All samples (BPNs, BPENs, BSA, and PUL) were freeze-dried and fully mixed with 200 mg of KBr and pelletized [26]. Spectrum scanning was taken in the wavelength range of 4000–400 cm^−1^, at a resolution of 4 cm^−1^, and with a scan speed of 2 mm/s, using a Fourier transform infrared spectrometer (FTIR, VERTEX70, Bruker, Germany).

#### 2.3.9. Morphological Observation of BPNs and BPENs

The freshly prepared samples were dropped onto the copper mesh, and the morphology of the BPNs and BPENs were observed using a transmission electron microscope (TEM, TECNAI F20, FEI Company, HIO, OSU, Columbus, OH, USA) after drying.

#### 2.3.10. Statistical Analysis

All the tests were repeated in triplicate, and the data are expressed as the means ± SDs (standard deviations). A one-way analysis of variance (ANOVA) and significant difference tests were performed using SPSS 27.0 software. Significant differences were determined with an independent sample *t*-test or Duncan’s multiple-range tests, at *p* = 0.05. Mean differences were considered significant when *p* < 0.05.

## 3. Results and Discussion

### 3.1. Particle Size and Zeta-Potential of BPNs

The results of the particle size of the nanoparticles are summarized in Table 1. It can be seen from Table 1 that the pH level had a significant effect on the particle size of the BPNs. As the pH deviated from the isoelectric point, the particle size first decreased and then increased, probably because they were in a neutral environment [27], which led to some particle aggregation. In addition, as shown in Table 1, the different BSA/PUL ratios had a significant effect on the particle size of the BPNs. The particle size of the BPNs at each pH was the smallest when the BSA/PUL ratio was 1:1. As the BSA/PUL ratio increased, the particle size of the BPNs also increased. This can be mainly attributed to the decrease in the concentration of PUL, which led to the rapid nucleation of the BSA at different rates, which, in turn, resulted in the formation of larger and more uneven particles [27]. Therefore, the optimal ratio of BSA/PUL was 1:1.

The electrostatic repulsion between nanoparticles is an important factor in determining the stability of the particles. Zeta potential is an important indicator in characterizing the stability of a colloidal dispersion [28]. It represents the coefficient of change in the surface charge and its absolute value represents the stability. It can be seen from Figure 2 that the potential of a solution was negative when the pH was 4.5–7.0, and that the potential gradually decreased with the increase in the pH. With the increase in the PUL concentration, the potential was at its maximum at a 1:1 ratio of PUL/BSA. This was because the surface of the BSA was negatively charged, whereas the surface of the PUL was not charged [29]. The decrease in the zeta potential of all the BPNs was obvious as the pH level increased from 4.5 to 5.5. This was because the charge of the PUL and BSA was neutralized; therefore, the electrostatic repulsion decreased. The aggregation of the BSA and PUL was also proved by the trend of the particle size increasing (Table 1).

### 3.2. Turbidity Determination and Thermal Stability

Nanoparticles may be affected by various pH environments during the manufacturing process, in storage, and during their passage through the human digestive tract. Therefore, it is necessary to assess the stability of a drug delivery system under different pH conditions. In this mixed solution system, the OD_600_ (Figure 3A) of the unheated BPNs sample over the entire pH range was close to 0, and the naked-eye view of the solution was transparent and translucent. This means that BSA and PUL form soluble complexes at different ratios and under different pH conditions. In general, the BPNs were stable over a wide pH range, which indicates a broad application prospect.

Thermal treatment—an important step in food preservation and production—can influence the structural and functional properties of proteins, resulting in the denaturation of proteins by disrupting the forces that stabilize their natural conformation [30]. The single BSA solution and the BPN samples with different pH levels were heated, and the results are shown in Figure 3B. We can see that when the pH was 4.5 in the four ratios, the sample solution had slightly flocculent precipitates. When the pH was 5.0, the sample solution had obviously larger flocs. When the pH was 5.5, the sample solution presented a more obvious milky-white colloidal solution state, and no precipitated flocs were formed. This was attributed to the thermal denaturation of the protein [31]. However, at pH 6.0, 6.5, and 7.0, this behavior was not observed. It is possible that the nanoparticle structure formed by the protein and polysaccharide was more stable at these pH conditions, demonstrating that PUL could improve the anti-agglomeration ability of nanoparticles and ensure that they are uniformly dispersed [32]. The denaturation temperature of BSA was reported to be between 62–65 °C [33], at which point it had a weak electrostatic repulsion and its interior hydrophobic groups were exposed [32] which, in turn, led to the formation of BSA aggregates.

In Figure 3B, the sample solution with a pH of 6.0 presented a translucent milky-white colloidal solution state. The sample solutions with a pH of 6.5 and 7.0 were clear and transparent. As the pH gradually increased, the solution state gradually became clear. The reason for this was that the isoelectric point of BSA is approximately 4.7, and the solution was easy to coagulate and flocculate near the isoelectric point. The more it deviated from the isoelectric point, the more stable the solution was [34].

As shown in Figure 3, it was observed that no turbidity or aggregates were formed at the three pH values of 6.0, 6.5, and 7.0, and this was true in both the heated group and the unheated group. However, the solution formed a state of suspension without precipitation at a pH of 6.0, which was more stable than the complex system prepared at pH values of 4.5–5.5. Therefore, it was indicated that BSA and PUL basically completed the self-assembly and presented a uniform and stable colloidal solution system under the pH 6.5–7.0 condition. Based on their smaller particle size and higher thermal stability, the BPNs (1:1) prepared at pH 6.5 were selected for the encapsulation of EGCG, and a series of characterization analyses were carried out.

### 3.3. BPENs’ Encapsulation Efficiency and Loading Rate

The encapsulation efficiency and loading rate of the BPENs loaded with EGCG are shown in Figure 4. As can be seen from Figure 4A, the encapsulation efficiency of BPENs (0.5) was 99.67%. With the increase in the EGCG loading concentration, the encapsulation efficiency of the BPENs did not change significantly.

In Figure 4B, we can see that the loading rate of BPENs (0.5) was 17.4%, and the loading rate also changed significantly with the increase in the concentration of EGCG. The loading of BPENs (2.5) reached 53.4%, which was 36.0% higher than that of BPENs (0.5).

### 3.4. In Vitro Antioxidant Activities

DPPH, DPPH, ABTS, FRAP, and hydroxyl radical scavenging capacity assays were carried out to evaluate the in vitro antioxidant capacity of the BPENs that were loaded with various concentrations of EGCG, and the results are shown in Figure 5.

DPPH is a very stable and classic nitrogen-containing free radical. EGCG can react with a DPPH free radical to form a phenoxy free radical. This reaction is stable, and it is not easy to cause new reactions. An analysis of the ability to eliminate DPPH· can determine EGCG antioxidant activity [35]. As shown in Figure 5A, the DPPH radicals scavenging activity was, in general, positively correlated with the sample concentration. The radical scavenging activities of the BPENs and EGCG (free) were significantly higher than those of the BPNs at all concentrations, suggesting that EGCG is a much more effective antioxidant than proteins and polysaccharides. The DPPH scavenging activity increased significantly with the increase in the antioxidant concentration. The spherical structure of the BPENs enables them to have a high surface area, which increases the interaction between the DPPH radicals and EGCG, thereby enhancing the antioxidant activity [36]. In addition, improving the water solubility of EGCG by nanoencapsulation can significantly enhance the scavenging ability of DPPH radicals. However, the DPPH scavenging ability of EGCG was significantly higher than that of the BPENs. This may be accounted for by the loss of exposed hydroxyl groups, since hydroxyl groups make a significant contribution to superior antioxidant activity [37]. However, when the EGCG was added to the protein, a much slower decrease in its antioxidant activity was observed. This was attributed to the protective effect provided by the protein, which was in line with previous studies [38].

Figure 5B shows that the ABTS^+^ free radical scavenging activity of the free EGCG was not very different from the activities of the free EGCG and BPENs that were loaded with different concentrations of EGCG. This shows that the nanoparticles can effectively improve the stability of EGCG and maintain its antioxidant activity. In line with the ideas of Laura et al. [39], we concluded that hydroxyl radicals are the most active oxygen-containing radicals, so the ability of BPENs to scavenge ·OH is an important indicator of their antioxidant activity [40]. FRAP is measured by monitoring the change in Fe^3+^ to Fe^2+^, which results from the electrons in the reducing substances [41]. As shown in Figure 5C, D, the FRAP scavenging activity and the hydroxyl radicals scavenging activity were also, in general, positively correlated with the sample concentration. Moreover, the free radical scavenging activity of free EGCG was not significantly different from that of the BPENs. When the concentration was increased, the free radical scavenging activity was significantly improved. This was mainly attributed to the higher surface area of the BPENs and the water solubility of EGCG after nanoencapsulation. The experimental results of the DPPH radicals scavenging activity and the ABTS^+^ scavenging activity were consistent, and it was shown that nanoparticles can enhance or retain the antioxidant activity of catechins [42].

### 3.5. In Vitro Release Test

The in vitro release behavior of the BPENs loaded with different concentrations of EGCG was investigated at 37 °C, and the results are shown in Figure 6A. The results of the in vitro release properties of EGCG showed that the BPENs loaded with different concentrations of EGCG showed a sustainable release ability. Similar release behavior characteristics were also observed in the carriers of soybean protein/EGCG nanoparticles [43]. In BPENs (0.5), EGCG was slowly released during the first hour. It is most likely that the EGCG release corresponded to the EGCG being closer to the nanoparticle surface. After 1 h, a sudden release of EGCG was observed, which lasted until after the second hour. The slow and sustained release of EGCG may be due to the diffusion of the EGCG embedded within the cores of the BPENs.

In BPENs (1.0), BPENs (1.5), BPENs (2.0), and BPENs (2.5), the release rate and the total release amount of EGCG increased with the increase in the initial concentration of EGCG. During the whole process, the BPENs exhibited a rapid but limited release of EGCG. Within the first hour, EGCG was rapidly released from the nanoparticles. The amount of EGCG released by the BPENs increased significantly with the increase in the EGCG-loaded concentration. After 1 h, the release of EGCG had significantly slowed and reduced. This release of EGCG was thought to be due to the partial adsorption of EGCG near the surface of the BPENs. The entire release process lasted for more than 5 h. The cumulative release rate of BPENs (0.5) at 6 h reached 37.19%, while the cumulative release rate of BPENs (2.5) reached 89.04%. It indicated that the cumulative release rate of the BPENs was related to the EGCG-loaded concentration, and the higher the concentration, the better the release effect was.

### 3.6. FTIR Analysis

The molecular characterization of the BPNs and BPENs was carried out using FTIR spectra. This technique has been used to evaluate the chemical and conformational changes that occur when nanoparticles are formed or when they interact with other compounds through the slight shift in the characteristic bands of the spectral regions of amide I and amide II. Figure 6B shows the FTIR spectra of BSA, PUL, BPNs, and BPENs. In Figure 6B, the FTIR spectra of PUL had a broad absorption band, at 3316 cm^−1^, which was caused by the -OH stretching vibration. Furthermore, an absorption peak appeared near 2925 cm^−1^ and 1640 cm^−1^, which was caused by the -CH_2_ stretching vibration and the O-C-O bond stretching vibration connecting the glucose units [44]. In the FTIR spectra of BSA, 3285 cm^−1^ was related to amide A, which was related to the N-H stretching; 2957 cm^−1^ was related to amide B, which was associated with the N-H stretching of the NH^3+^ free ion; 1643 cm^−1^ was the peak of amide I and the C=O stretching vibration; the absorption peak at 1530 cm^−1^ was related to amide II, which was related to the C-N stretching and the N-H bending vibration; and 1392 cm^−1^ was related to amide III, which was associated with the C-N stretching and the N-H bending [45]. In the case of the BPNs, the presence of the protein enhanced the intensity of the 1643 and 1530 cm^−1^ bands, due to superposition with the amide I and amide II bands, respectively [46]. Furthermore, small shifts in the observation bands were observed, compared to BSA and the BPNs [45]. The spectral bands of the BPNs showed the characteristic peaks of these proteins and polysaccharides, which indicates that BSA and PUL produced BPNs through noncovalent binding, which is the self-assembly effect. A comparison of the FTIR spectrum of the BPENs and BPNs showed that both have similar profiles, suggesting that the structure of the nanoparticles does not change significantly with the addition of EGCG.

In the Fourier transform infrared spectra of the BPENs, 1284 cm^−1^ was related to the stretching vibration of the C-O bond of EGCG. Therefore, the BPENs had one more absorption peak as a result of the stretching vibration of the C-O bond of the catechins, compared with the BPNs. This indicates that the nanoparticles successfully embedded in the EGCG. Meanwhile, the presence of the aromatic ring quadrant, the -OH deformation of the aromatic alcohols, and the C-O stretching of the aromatic alcohols and aliphatic diols were shown at 1078 cm^−1^ [47]. This indicates that no bonds were formed between the BPNs and EGCG.

### 3.7. TEM Observation of Microscopic Morphology

TEM images can be used to directly observe the morphology and structure of samples. As shown in Figure 7A, it can be seen that the BPNs (BSA: PUL = 1:1 and pH = 6.5) are spherical and that the particle size is approximately 350 nm, which is roughly the same as the results measured by the particle size analyzer. The particle morphology was similar to the shape of woolen balls. The inner and outer contrasts were different, indicating that PUL and BSA self-assembled to form spherical nanoparticles. The images demonstrate a spherical morphology for the nanoparticles, which corroborates the literature that has been published [48].

According to Figure 7B, it can be seen that the shape of the EGCG-encapsulated composite particles changed, indicating that EGCG might be successfully encapsulated inside the BPNs.

## 4. Conclusions

In conclusion, EGCG-loaded BSA/PUL nanoparticles were successfully fabricated via self-assembly. This was driven by hydrogen bonding, which provided the desirable EGCG-loading efficiency, high stability, and a strong antioxidant capacity. The encapsulation efficiency of the BPENs was above 99.0%. The BPENs had high antioxidant activity in vitro, and their antioxidant capacity increased with the increase in the EGCG concentration. The in vitro release assays showed that the BPENs were released continuously over 6 h. An FTIR analysis indicated the presence of hydrogen bonding, hydrophobic interactions, and electrostatic interactions, which were the driving force for the formation of the EGCG carrier nanoparticles. The TEM images demonstrated that the BPNs and BPENs both exhibited regular spherical particles. In conclusion, BPENs are good delivery carriers for enhancing the stability and the antioxidant activity of EGCG, which is of great importance in the development of functional foods.

## Figures and Tables

**Figure 1 foods-11-04074-f001:**
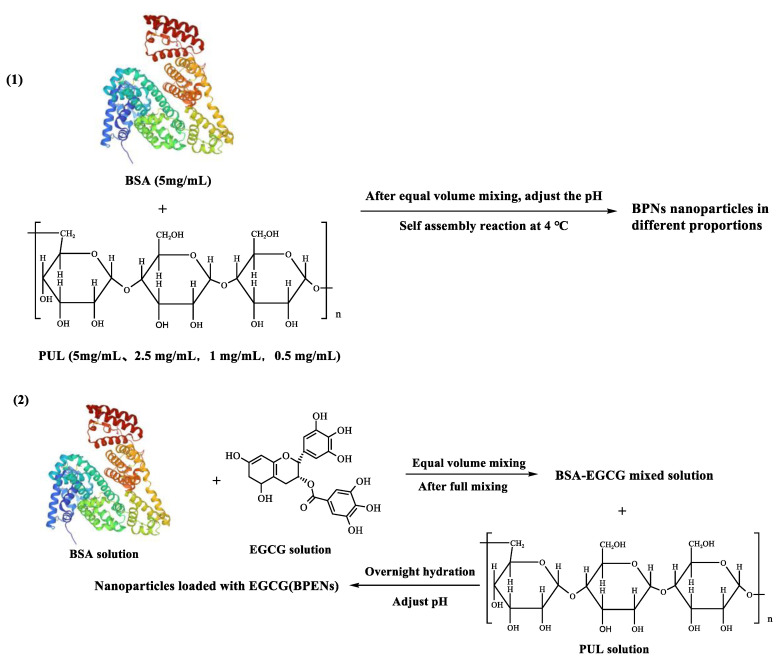
The preparation process of BPNs (**1**) and BPENs (**2**).

**Figure 2 foods-11-04074-f002:**
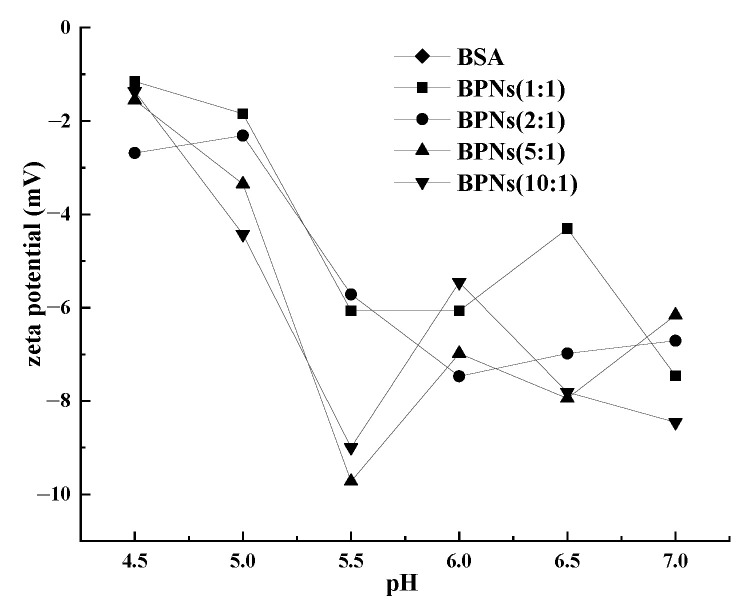
Zeta potential of BPNs as a function of pH.

**Figure 3 foods-11-04074-f003:**
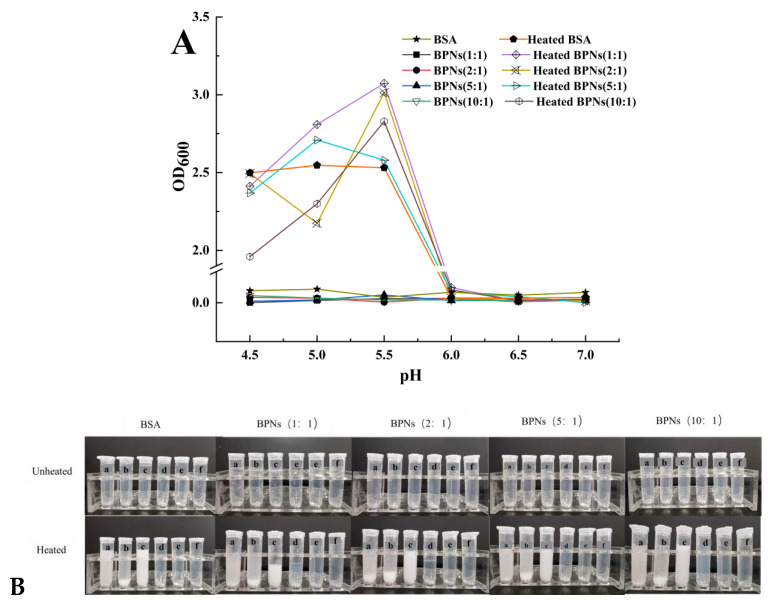
Turbidity curves (**A**) and visual appearance (**B**) of BSA and BPNs as a function of pH, without being heated and heated at 80 °C for 30 min. a–f represents pH 4.5, 5.0, 5.5, 6.0, 6.5, and 7.0, respectively.

**Figure 4 foods-11-04074-f004:**
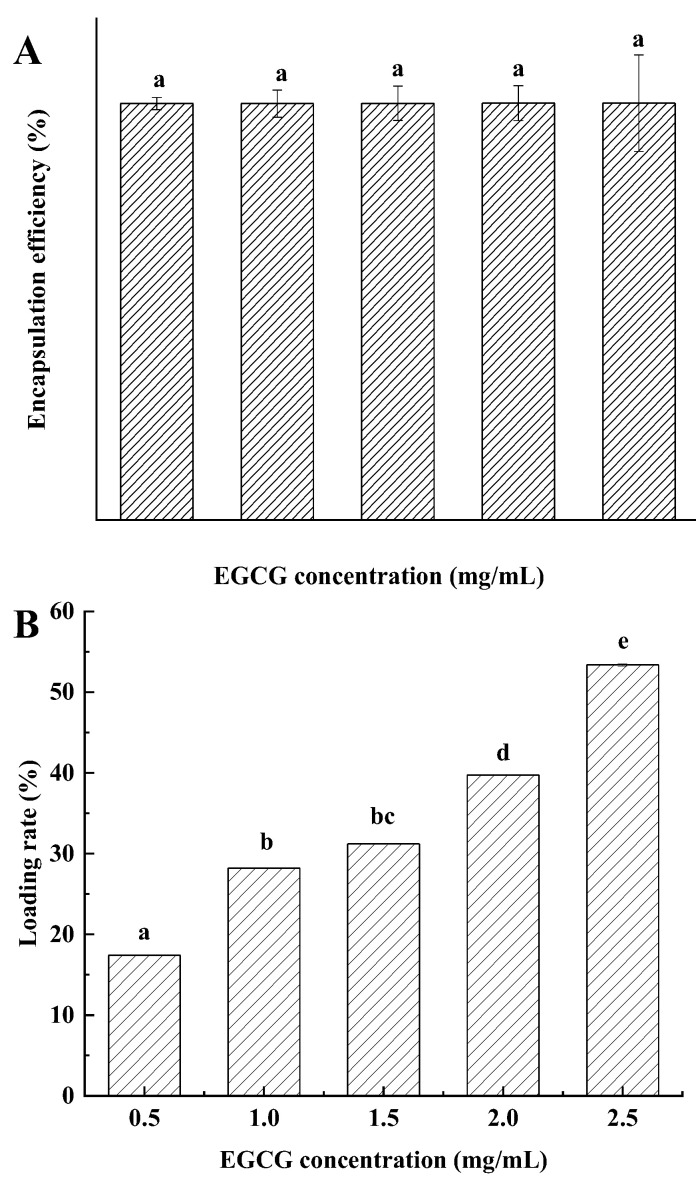
Encapsulation efficiency (**A**) and loading rate (**B**) of BPENs. Histograms followed by different letters were significantly different at *p* < 0.05.

**Figure 5 foods-11-04074-f005:**
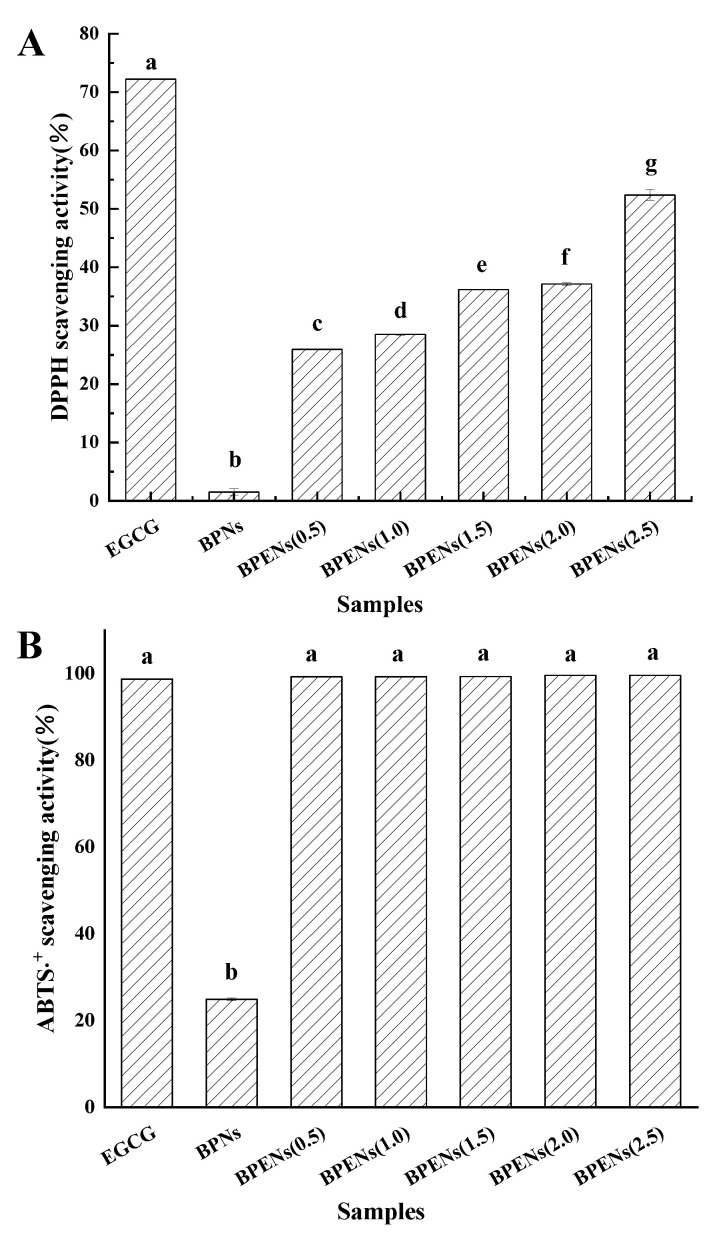
The in vitro antioxidant capacity of BPENs, as determined by the DPPH (**A**), ABTS (**B**), FRAP (**C**), and hydroxyl radical scavenging capacity (**D**), respectively. Histograms followed by different letters were significantly different at *p* < 0.05.

**Figure 6 foods-11-04074-f006:**
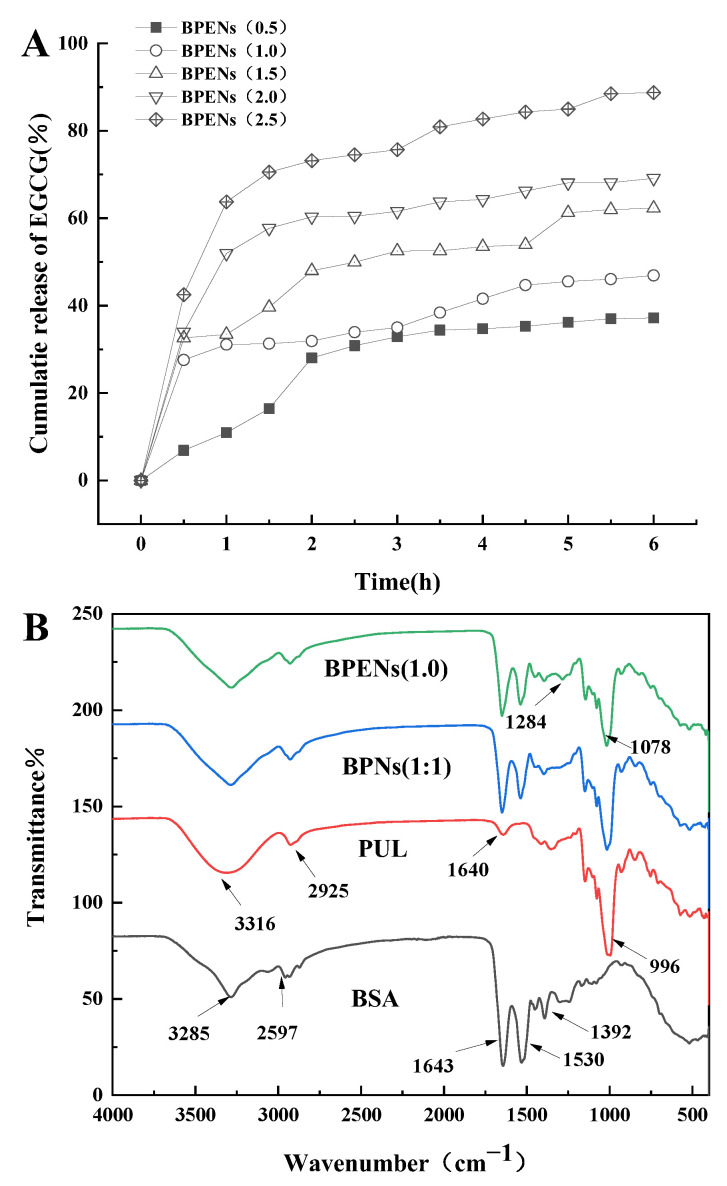
In vitro release curves (**A**) of EGCG and FTIR spectrum (**B**) of BSA, PUL, BPNs, BPENs.

**Figure 7 foods-11-04074-f007:**
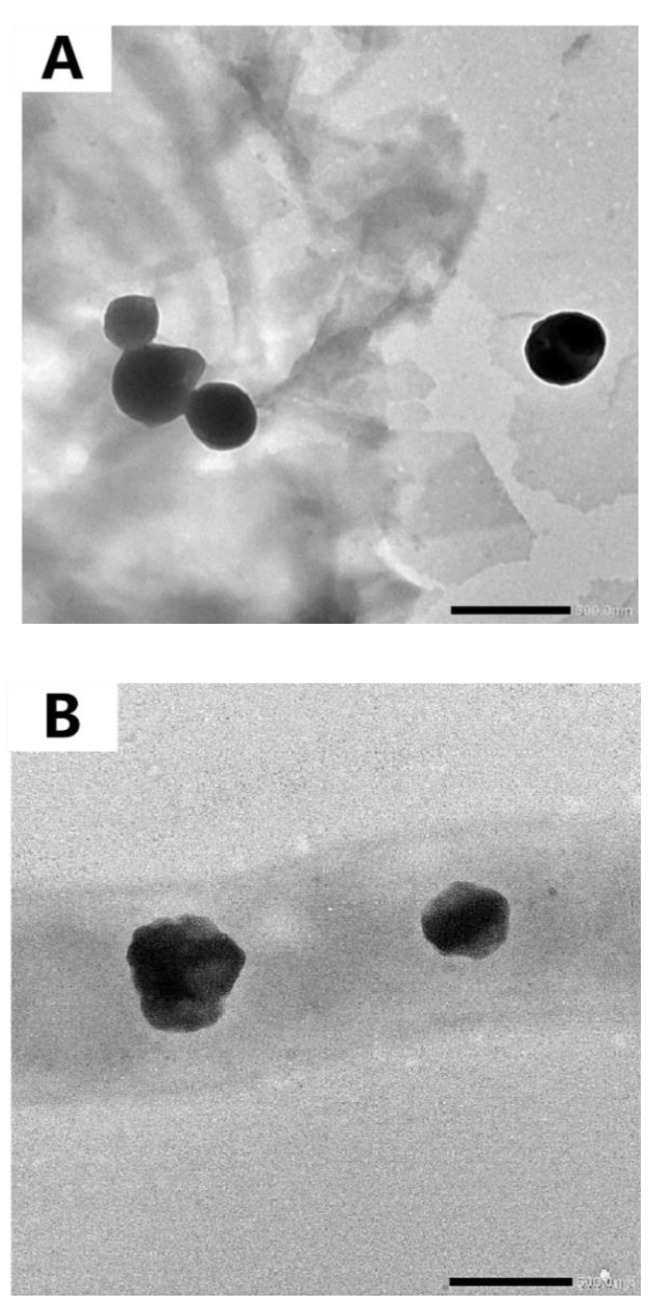
TEM images of BPNs (1:1) (**A**) and BPENs (1.0) (**B**).

**Table 1 foods-11-04074-t001:** Particle size and polymer dispersity index (PDI) of BPNs.

pH	1:1		2:1		5:1		10:1	
	Particle Size (nm)	PDI	Particle Size (nm)	PDI	Particle Size (nm)	PDI	Particle Size (nm)	PDI
4.5	341 ± 5 ^a^	0.369	382 ± 5 ^a^	0.516	558 ± 8 ^a^	0.521	462 ± 45 ^a^	0.375
5.0	219 ± 6 ^b^	0.392	335 ± 26 ^b^	0.372	325 ± 9 ^b^	0.350	644 ± 48 ^b^	0.330
5.5	180 ± 8 ^c^	0.369	226 ± 1 ^c^	0.218	251 ± 21 ^c^	0.122	620 ± 57 ^bc^	0.312
6.0	139 ± 13 ^d^	0.355	447 ± 32 ^d^	0.341	606 ± 56 ^ad^	0.300	224 ± 21 ^d^	0.321
6.5	251 ± 11 ^e^	0.265	488 ± 37 ^de^	0.190	670 ± 3 ^e^	0.402	257 ± 13 ^de^	0.434
7.0	347 ± 28 ^af^	0.449	362 ± 18 ^abf^	0.546	2011 ± 62 ^f^	0.503	749 ± 50 ^f^	0.303

Values that do not bear the same letter in the same column were significantly different (*p* < 0.05).

## Data Availability

The data are contained within the article.

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
