# Peer review of "The Loading of Epigallocatechin Gallate on Bovine Serum Albumin and Pullulan-Based Nanoparticles as Effective Antioxidant"

_foods, 2022, doi:10.3390/foods11244074_

Round 1
Reviewer 1 Report
This study is significant and justified with the comprehensive set of techniques. Although I have a few remarks on the scientific aspects of this paper, the English language should be significantly improved.
From my point of view, the following concerns should be addressed (including improvement of English):
Line 26: The introduction part should be started with sentences highlighting the biological significance of catechins. After that, authors could emphasize the chemical structure of EGCG or catechins in general.
Line 35: Sentence "Thus, EGCG is a natural, preservative and bacteriostatic raw material" should be rewritten. EGCG is a molecule, not raw material.
Line 45: This expression is strange. Please, re-phrase or remove it.
Line 56: BSA could be allergenic
Line 58: There are no nine hydroxyl groups in the glucose unit. Please, rewrite
Line 94: I would say incubated instead of hydrated
Line 120: Please, state the centrifugal force in g units; it is more relevant than RPM.
Line 190: nm or cm-1? I think it should be cm-1.
Line 211. The word figure is redundant.
Lines 220, 246, 247, 250, 251: Pullulan per se does not have the charged groups. Did the authors modify the structure of pullulan by introducing the charged groups? Please clarify.
Line 238 (Figure 2): According to particle size distribution (Figure 2), BSA size is between 10000 and 100 nm, far from the BSA monomer's size. The authors should clarify this point. Is the BSA in aggregated form in this study? Furthermore, why did the authors not present the size distribution results at pH values below 6 in Figure 2? On the other hand, these results are shown in Table 1.
Line 287: In Figure 3, the authors should clearly state what the letters A-F mean. Different pH values?
Lines 370-371: Based on in vitro release test, how did authors conclude that BPEN nanoparticles protect EGCG?
Line 396 and Figure 6: The difference between FTIR spectra of BPN and BPEN is not obvious. The authors should zoom the spectral region around 1284 cm-1.
Author Response
Comment: This study is significant and justified with the comprehensive set of techniques. Although I have a few remarks on the scientific aspects of this paper, the English language should be significantly improved.
Answers: We appreciate the selfless contributions to this manuscript from you. It is because of your insightful and helpful comments and suggestions, we gained the confidence to improve our work better. And as suggested by the referee, the manuscript has been edited for language using editing service of MDPI.
Comment 1: The introduction part should be started with sentences highlighting the biological significance of catechins. After that, authors could emphasize the chemical structure of EGCG or catechins in general. And the Sentence "Thus, EGCG is a natural, preservative and bacteriostatic raw material" should be rewritten. EGCG is a molecule, not raw material.
Answers: Thank you very much for your comments and additions. We have revised the content and deleted inappropriate sentences; the revised contents are as follows: “Previous studies have shown that catechins have antioxidant effects [1] and that they are highly effective natural antioxidants, with low toxicity [2]. Furthermore, they also have anti-cancer [3], anti-obesity [4], and anti-inflammatory properties [5]. Catechins contain three basic ring nuclei-A, B and C-which form a D ring after esterification. According to the different groups connected to the B and C rings, common catechins can be divided into four forms, which can be divided as follows: epicatechin (EC), epigallocatechin (EGC), epicatechin gallate (ECG), and epigallocatechin gallate (EGCG) [6]. Among them, EGCG has the highest content, accounting for approximately 70% of the total catechins. The molecular structure of EGCG contains eight monophenolic hydroxyl groups and it is the most active catechin.” in lines 29-38 in the revised manuscript.
And the sentence "Thus, EGCG is a natural, preservative and bacteriostatic raw material" mentioned by the referee was deleted after rewriting of the section.
Comment 2: Line 45: This expression is strange. Please, re-phrase or remove it.
Answers: Many thanks for your comment. We have revised the sentence "However, the solubility of EGCG both in the aqueous and oil phases is poor. " into " However, the solubility of EGCG is poor." in line 45 in the revised abstract.
Comment 3: Line 56: BSA could be allergenic
Answers: Thank you very much for your comments. We have revised the sentence and supplemented references “BSA is widely used as a drug carrier due to its non-antigenicity, its degradability, its non-toxicity [12], in spite of and its allergenic disadvantage [13].” in lines 59-60 in the revised manuscript
- Rostamnezhad, F., Hossein Fatemi, M. Comprehensive investigation of binding of some polycyclic aromatic hydrocarbons with bovine serum albumin: Spectroscopic and molecular docking studies. Bioorg Chem 2022, 120, 105656, https://doi.org/10.1016/j.ultsonch.2019.104678.
Comment 4: Line 58: There are no nine hydroxyl groups in the glucose unit. Please, rewrite
Answers: Thank you for your comments. We are sorry for our careless mistake and have corrected it in line 62 in the revised manuscript. The revised sentence is as follows: “PUL contains three hydroxyl groups in each glucose unit and has good solubility in aqueous solution.”
Comment 5: Line 94: I would say incubated instead of hydrated
Answers: Thank you for your comments. We have replaced “hydrated” with “incubated” in line 99 in the revised manuscript. The revised sentence is as follows: “Then, the solution was incubated overnight to make it completely mixed and homogeneous.”
Comment 6: Line 120: Please, state the centrifugal force in g units; it is more relevant than RPM.
Answers: Thank you for your comments. We have replaced rpm/min with g in line 126 in the revised manuscript. The revised sentence is as follows: “The BPENs solutions were centrifuged at 3500×g for 30 min.”
Comment 7: Line 190: nm or cm-1? I think it should be cm-1
Answers: Many thanks for your comments. We are sorry for our careless mistake and have replaced nm with cm-1 in line 197 in the revised manuscript. The revised sentence is as follows: “Spectrum scanning was taken in the wavelength range of 4000-400 cm-1 at a resolution of 4 cm-1 with scan speed of 2 mm/s using Fourier transform infrared spectrometer (FTIR, VERTEX70, Bruker, Germany).”
Comment 8: Line 211. The word figure is redundant.
Answers: Thank you for your comments. We have deleted the word “figure” and rearranged this section in line 215 in the revised manuscript. The revised sentence is as follows: “The results of the particle size determination of the nanoparticles are summarized in Table 1.”
Comment 9: Lines 220, 246, 247, 250, 251: Pullulan per se does not have the charged groups. Did the authors modify the structure of pullulan by introducing the charged groups? Please clarify
Answers: Many thanks for the comment. In combination with the comments of other reviewers, we have rewritten these sections. The modifications are as follows: " It can be seen from Table 1 that the pH level had a significant effect on the particle size of the BPNs. As the pH deviated from the isoelectric point, the particle size first decreased and then increased-probably because they were in a neutral environment [27]-which led to some particle aggregation. In addition, as shown in Table 1, the dif-ferent BSA/PUL ratios had a significant effect on the particle size of the BPNs. The par-ticle size of the BPNs at each pH was the smallest when the BSA/PUL ratio was 1:1. As the BSA/PUL ratio increased, the particle size of the BPNs also increased. This can be attributed, mainly, to the decrease in the concentration of PUL, which led to the rapid nucleation of the BSA at different rates, which, in turn, resulted in the formation of larger and more uneven particles [27]. Therefore, the optimal ratio of BSA/PUL was 1:1" in lines 221-230 in the revised manuscript.
" Nanoparticles may be affected by various pH environments during the manufacturing process, in storage, and during their passage through the human digestive tract. Therefore, it is necessary to assess the stability of a drug delivery system under different pH conditions. In this mixed solution system, the OD600(Fig. 3A) of the unheated BPNs sample over the entire pH range is close to 0, and the naked-eye view of the solution is transparent and translucent. This means that BSA and PUL form soluble complexes at different ratios and under different pH conditions. In general, the BPNs were stable over a wide pH range, which indicates a broad application prospect" in lines 244-252 in the revised manuscript.
Comment 11: Line 238 (Figure 2): According to particle size distribution (Figure 2), BSA size is between 10000 and 100 nm, far from the BSA monomer's size. The authors should clarify this point. Is the BSA in aggregated form in this study? Furthermore, why did the authors not present the size distribution results at pH values below 6 in Figure 2? On the other hand, these results are shown in Table 1
Answers: Many thanks for the referee’s comment. As the decrease in pH will weaken the intermolecular electrostatic repulsion of BSA, it will increase the possibility of protein aggregation, resulting in the particle size of BSA at pH 6.0 is significantly greater than the BSA monomer's size. We have determined the particle size distribution of BPNs at all pH, but it is not shown in the article. We have deleted it according to your suggestion. Please refer to Table 1 for the specific data of particle size.
Comment 12: In Figure 3, the authors should clearly state what the letters A-F mean. Different pH values?
Answers: Many thanks for your comments. We have revised the letters A-F into a-f in the Fig. 3(B), and added explanation " a-f represent pH 4.5, 5.0, 5.5, 6.0, 6.5, and 7.0, respectively." in line 294 in the revised manuscript.
Comment 13: Lines 370-371: Based on in vitro release test, how did authors conclude that BPENs nanoparticles protect EGCG?
Answers: Many thanks for the referee’s comment. The conclusion that "BPENs nanoparticles protect EGCG" is slightly inappropriate from the results of this experiment. Therefore, we have deleted inaccurate sentence in line 380 in the revised manuscript.
Comment 14: Line 396 and Figure 6: The difference between FTIR spectra of BPN and BPEN is not obvious. The authors should zoom the spectral region around 1284 cm-1.
Answers: Many thanks for the referee’s comment. We are sorry that the 1284 cm-1 peak is incorrectly marked due to our carelessness. We have corrected it in the revised Fig.6(B).

Reviewer 2 Report
In general, this article provides interesting and innovative results in the field of nanoencapsulation, however, in the objective it is not clear in which system these nanocapsules could be applied.

Author Response
Comment: In general, this article provides interesting and innovative results in the field of nanoencapsulation, however, in the objective it is not clear in which system these nano capsules could be applied.
Answers: We appreciate the selfless contributions to this manuscript from you. It is because of your insightful and helpful comments and suggestions, we gained the confidence to improve our work better. BPENs are a good delivery carrier for enhancing the stability and antioxidant activity of EGCG, which is of great importance in the development of functional foods.
Comment 1: BPNs: control? nanoparticles with bovine serum albumin+pulluan?
Answers: Thank you for your comments. We have added the abbreviation explanation about BPNs which refers to BSA/PUL-based nanoparticles in line 23 in the revised abstract.
Comment 2: The article "low-toxic natural antioxidants" is not supported by the literature.
Answers: Thank you for your comments. Now reference is included in the revised manuscript. Specific reference is listed as follows: “Studies have shown that catechin have antioxidant effects [1], which are highly effective and low-toxic natural antioxidants [2].”
- Song, H.; Wang, Q.; He, A.; Li, S.; Guan, X.; Hu, Y.; Feng, S. Antioxidant activity, storage stability and in vitro release of epigallocatechin-3-gallate (EGCG) encapsulated in hordein nanoparticles. Food Chemistry 2022, 388, 132903, doi:https://doi.org/10.1016/j.foodchem.2022.132903.
Comment 3: In this context, an attempt was made to encapsulate EGCG into a BSA-PUL nanoparticulate (BPNs) system to retain or enhance their antioxidant potential. And release capacity??in foods?? What type of active release is intended to be obtained and why, for example, rapid, controlled?
Answers: Thank you for your comments. This article focused at the effect of the ratio of BSA and PUL and the preparation pH on the particle size, polydispersity index (PDI), zeta potential and thermal stability of BSA or BSA-PUL complexes to gain better nanoparticles for EGCG loading. And the effect of EGCG concentrations on the encapsulation efficiency, loading rate, antioxidant activity and release capacity in vitro of BPENs were determined. However, the type of release behavior was not further studied. We cannot gain corresponding conclusions. Therefore, we did not mention the type of release behavior.
Comment 4: The sentence "Pullulan (PUL) is a natural degradable macromolecular polymer, which is non-hazardous and has been widely used in many fields." is not supported by the literature.
Answers: Thank you for your comments. Now reference is included in the revised manuscript. Specific reference is listed as follows: “Pullulan (PUL) is a natural degradable macromolecular polymer, which is non-hazardous and has been widely used in many fields [14].”
- Roy, S.; Rhim, J.-W. Effect of chitosan modified halloysite on the physical and functional properties of pullulan/chitosan biofilm integrated with ruin. Applied Clay Science 2021, 211, 106205, doi:https://doi.org/10.1016/j.clay.2021.106205.
Comment 5: "Finally, the ratios of BSA and PUL were 1:1, 2:1, 5:1, and 10:1 (all of the BSA concen-trations were 2.5 mg/mL). "PUL/BSA? 1:1 2:1 5:1 10:1?All of the BSA concentrations were 2.5 mg/ml?
Answers: Thank you for your comments. We have replaced "BSA and PUL" with "BSA/PUL" in lines 92-94 in the revised manuscript. The revised sentence is as follows: “Equal volumes of 5 mg/mL BSA solution and the above four concentrations of PUL (5 mg/mL, 2.5 mg/mL, 1 mg/mL, and 0.5 mg/mL, respectively.) were mixed, respectively. Finally, the ratios of BSA/PUL were 1:1, 2:1, 5:1, and 10:1 (all of the BSA concentrations were 2.5 mg/mL).” In other words, we kept the concentration of BSA the same in all samples, adjusting by the concentration of PUL to obtain different proportions of samples.
Comment 6: Why 80℃?
Answers: Thank you for your comments. We have referred to the 80 ℃ finally determined in some literatures. We have quoted the article. Please refer to lines 105-106 in the revised manuscript. The sentence is as follows: “The BSA and BPNs solutions with different ratios and pH levels were heated at 80°C for 30 min and then cooled to room temperature in ice water [19].”
- Zhu, L.; Yang, F.; Li, D.; Wu, G.; Zhang, H. Preparation, structure and stability of protein-pterostilbene nanocomplexes coated by soybean polysaccharide and maltodextrin. Food Bioscience 2022, 49, 101899, doi:https://doi.org/10.1016/j.fbio.2022.101899.
Comment 7: concentretion was spelling errors.
Answers: Thank you for your comments. We are sorry for our careless mistake and have corrected the “concentration” into “concentration” in revised Figure S1 in the revised manuscript.
Comment 8: The particle size first decreased and then increased. It is mainly attributed to the enhanced electrostatic interaction between BSA and PUL through the change of their charge density. Add discussion with literature?? As the concentration of PUL gradually decreased, the particle size and PDI also increased gradually, which was due to the aggregation of BSA with a larger concentration. In PDI, but particle size with 2:1, decreased.
Answers: Many thanks for the comment. These parts of the language are really inappropriate. In combination with the comments of other reviewers, we have rewritten these sections. The modifications are as follows: " It can be seen from Table 1 that the pH level had a significant effect on the particle size of the BPNs. As the pH deviated from the isoelectric point, the particle size first decreased and then increased-probably because they were in a neutral environment [27]-which led to some particle aggregation. In addition, as shown in Table 1, the different BSA/PUL ratios had a significant effect on the particle size of the BPNs. The particle size of the BPNs at each pH was the smallest when the BSA/PUL ratio was 1:1. As the BSA/PUL ratio increased, the particle size of the BPNs also increased. This can be attributed, mainly, to the decrease in the concentration of PUL, which led to the rapid nucleation of the BSA at different rates, which, in turn, resulted in the formation of larger and more uneven particles [27]. Therefore, the optimal ratio of BSA/PUL was 1:1" in lines 221-230 in the revised manuscript.
- Lu, J.; Xie, L.; Wu, A.; Wang, X.; Liang, Y.; Dai, X.; Cao, Y.; Li, X. Delivery of silybin using a zein-pullulan nanocomplex: Fabrication, characterization, in vitro release properties and antioxidant capacity. Colloids and Surfaces B: Biointerfaces 2022, 217, 112682, doi:https://doi.org/10.1016/j.colsurfb.2022.112682.
Comment 9: With the increase of PUL concentration, the potential was maximum when the BSA-PUL concentration ratio was 1:1. At 1:1 ratio of PUL/BSA are equal.
Answers: Thank you for your comments. We have revised the sentence “With the increase of PUL concentration, the potential was maximum when the BSA-PUL concentration ratio was 1:1.” into “With the increase in the PUL concentration, the potential was at its maximum at a 1:1 ratio of PUL/BSA” in lines 236-237 in the revised manuscript.
Comment 10: Add abbreviation explanation of PDI
Answers: Thank you for your comments. We have added abbreviation explanation of PDI in line 245 in the revised manuscript. The revised sentence is as follows: “Table 1. Particle size, polydispersity index (PDI) of BPNs.”
Comment 11: At pH 6.0? In Fig 3A, the samples without and with heating cannot be correctly differentiated
Answers: Thank you for your comments. We have adjusted the color and pattern of the lines in Fig. 3(A) to make it easier to distinguish.
In addition, we have rewritten the sentence that the reviewer mentioned which may make the reader confused, to improve its readability. The modifications are as follows:
"Nanoparticles may be affected by various pH environments during the manufacturing process, in storage, and during their passage through the human digestive tract. Therefore, it is necessary to assess the stability of a drug delivery system under different pH conditions. In this mixed solution system, the OD600(Fig. 3A) of the unheated BPNs sample over the entire pH range is close to 0, and the naked-eye view of the solution is transparent and translucent. This means that BSA and PUL form soluble complexes at different ratios and under different pH conditions. In general, the BPNs were stable over a wide pH range, which indicates a broad application prospect" in lines 248-255 in the revised manuscript.
Comment 12: This has been attributed to thermal denaturation of the protein [26]. But in this case at low pH, because at pH 6.0, 6.5 and 7.0, this behavior is not observed.
Answers: Thank you for your comments. We have rewritten the sentence as follow: "However, at pH 6.0, 6.5, and 7.0, this behavior was not observed. It is possible that the nanoparticle structure formed by the protein and polysaccharide was more stable at these pH conditions, demonstrating that PUL could improve the anti-agglomeration ability of nanoparticles and ensure that they are uniformly dispersed [32]. The denaturation temperature of BSA was reported to be between 62-65°C [33], at which point it had a weak electrostatic repulsion and its interior hydrophobic groups were exposed [32], which, in turn, led to the formation of BSA aggregates." in lines 264-268 in the revised manuscript.
- de Oliveira, R.C.; Benevides, C.A.; Rodrigues, G.C.P.; Tenório, R.P. Thermal Denaturation and γ-Irradiation effects on the Crack Patterns of Bovine Serum Albumin (BSA) Dry Droplets. Colloid and Interface Science Communications 2019, 28, 15-19, doi:https://doi.org/10.1016/j.colcom.2018.11.003.
Comment 13: Based on the particle size distribution and thermal stability analysis, BPNs (1:1) prepared at pH 6.5 were selected for the encapsulation of EGCG and a series of characterization analyses were carried out. In relation to this aspect, what was considered?
Answers: Thank you for your comments. We have drawn this conclusion according to the particle size of BPNs and the thermal stability analysis. According to the results of thermal stability, the heating group and unheated group did not aggregate at pH 6.5, and 7.0 and showed higher stability than others. According to Table 1, the particle size of BPN is the smallest when the BSA/PUL ratio is 1:1. Compared with BSA/PUL (1:1) prepared at pH 7.0, BSA/PUL (1:1) prepared at pH 6.5 showed smaller particle size. Therefore, BPNs (1:1) prepared at pH 6.5 were selected for the encapsulation of EGCG and a series of characterization analyses were carried out.
Thus, we have rewritten the sentence as follow: “Based on smaller particle size and higher thermal stability, BPNs (1:1) prepared at pH 6.5 were selected for the encapsulation of EGCG and a series of characterization analyses were carried out.”
Comment 14: Significantly different markers
Answers: Thank you for your comments. We have modified the marking of significant differences in revised Fig.4A and Fig.5B in the revised manuscript.
Comment 15: The DPPH scavenging activity increased significantly with increasing antioxidant concentration. The spherical structure of BPENs nanoparticles enables them to have a high surface area which increases the interaction between DPPH radicals and EGCG, thereby enhancing the antioxidant activity. There are similar results in other studies in the literature.
Answers: Thank you for your comments. Reference is included in the revised manuscript. Specific reference is listed as follows: “The DPPH scavenging activity increased significantly with the increase in the anti-oxidant concentration. The spherical structure of the BPENs enables them to have a high surface area, which increases the interaction between the DPPH radicals and EGCG, thereby, enhancing the antioxidant activity [36].” in line 323 in the revised manuscript.
- Fan, Y.; Liu, Y.; Gao, L.; Zhang, Y.; Yi, J. Improved chemical stability and cellular antioxidant activity of resveratrol in zein nanoparticle with bovine serum albumin-caffeic acid conjugate. Food Chemistry 2018, 261, 283-291, doi:https://doi.org/10.1016/j.foodchem.2018.04.055.
Comment 16: In addition, the presence of PUL molecules increases the viscosity of the reaction system, which is not conducive to the interaction between DPPH radicals and EGCG. It's not understood.
Answers: Thank you for your comments. We found this sentence slightly inappropriate and have deleted it in lines 331-333 in the revised manuscript.
Comment 17: Based on the above analysis, BPENs (1.0) are finally selected for structural and TEM analysis. Why?
Answers: Thank you for your comments. In the process of FTIR analysis and TEM analysis, we did not consider many other factors, but chose BPENs with moderate loading rate for the next analysis. Considering that this sentence is inappropriate, we have deleted it in lines 381-382 in the revised manuscript.
Comment 18: TEM images demonstrated that the BPNs and BPENs both exhibited irregular spherical particles. Irregular? The particle morphology was similar to the shape of woolen balls??
Answers: Thank you for your comments. We have modified it according to your comments, and changed the “irregular” into “regular” in line 444 in the revised manuscript. The revised sentence is as follows: “The TEM images demonstrated that the BPNs and BPENs both exhibited regular spherical particles”
Comment 19: In conclusion, BPENs nanoparticles are a good delivery carrier for enhancing the stability and antioxidant activity of EGCG. According to the results, what would be its potential application?
Answers: Thank you for your comments. We have revised the sentence:“In conclusion, BPENs nanoparticles are a good delivery carrier for enhancing the stability and antioxidant activity of EGCG.” into “In conclusion, BPENs nanoparticles are a good delivery carrier for enhancing the stability and antioxidant activity of EGCG, which is of great importance in the development of functional foods.” in line 446 in the revised manuscript.

Reviewer 3 Report
The study is a very interesting and current topic. The authors treated the subject with the same attractiveness.
Manuscript is written in a very simple and clear language. The experiments were explained clearly and clearly and the discussion was done at a sufficient level.
Minor spelling and writing errors are noted in the attached file.

Author Response
Comment: The study is a very interesting and current topic. The authors treated the subject with the same attractiveness. Manuscript is written in a very simple and clear language. The experiments were explained clearly and clearly and the discussion was done at a sufficient level. Minor spelling and writing errors are noted in the attached file.
Answers: We appreciate the selfless contributions to this manuscript from you. It is because of your insightful and helpful comments and suggestions, we gained the confidence to improve our work better. We have been carefully corrected these mistakes and revised with the revisions marked in red. The point-by-point answers to these comments and suggestions were listed as below.
Comment 1: Besides, it also has anti-cancer, anti-cancer, anti-obesity action and anti-inflammatory properties. The word anti-cancer is written twice.
Answers: Thank you for your comments. We are sorry for our careless mistake and have deleted anti-cancer in lines 30-31 in the revised manuscript. The revised sentence is as follows: Furthermore, they also have anti-cancer [3], anti-obesity [4], and anti-inflammatory properties [5].”
Comment 2: Equal volumes of 5 mg/mL BSA solution and the above four concentrations of PUL were respectively mixed. Finally, the ratios of BSA and PUL were 1:1, 2:1, 5:1, and 10:1 (all of the BSA concentrations were 2.5 mg/mL). Please give details instead of above.
Answers: Thank you very much for your comments and additions. We have supplemented some content to clarify the concentration of PUL in line 95 in the revised manuscript. The revised sentence is as follows: “Equal volumes of 5 mg/mL BSA solution and the above four concentrations of PUL solutions (5 mg/mL, 2.5 mg/mL, 1 mg/mL, and 0.5 mg/mL, respectively.) were mixed, respectively.”
